# The Role of Telemedicine in Children with Obstructive Sleep Apnea Syndrome (OSAS): A Review of the Literature

**DOI:** 10.3390/jcm13072108

**Published:** 2024-04-04

**Authors:** Luisa Rizzo, Elena Barbetta, Flaminia Ruberti, Matilde Petz, Marco Tornesello, Michela Deolmi, Valentina Fainardi, Susanna Esposito

**Affiliations:** Pediatric Clinic, Department of Medicine and Surgery, University of Parma, 43126 Parma, Italy; luisa.rizzo811@gmail.com (L.R.); elena.barbetta@unipr.it (E.B.); flaminia.ruberti@gmail.com (F.R.); matilde.petz@unipr.it (M.P.); marco.tornesello@unipr.it (M.T.); valentina.fainardi@unipr.it (V.F.)

**Keywords:** obstructive sleep apnea syndrome, polysomnography, telemedicine, telediagnosis, teletherapy, telemonitoring

## Abstract

The advent of telemedicine marked a significant turning point in the healthcare landscape, introducing a revolutionary approach to the delivery of medical care. Digital technologies easily connect health professionals and patients, overcoming geographical and temporal barriers. Telemedicine has been used for sleep disorders including obstructive sleep apnea syndrome (OSAS) since the mid-1990s. In adult patients with OSAS, telemedicine is helpful both for consultation and diagnosis, the latter obtained through remote recordings of oxygen saturation and further parameters registered with telemonitored respiratory polygraphy or polysomnography. Remote monitoring can be used to follow up the patient and verify adherence to daily treatments including continuous positive airway pressure (CPAP). In children, studies on the role of telemedicine in OSAS are scarce. This narrative review aims to describe the application of telemedicine in children with obstructive sleep apnea syndrome (OSAS), assessing its advantages and disadvantages. In patients with OSA, telemedicine is applicable at every stage of patient management, from diagnosis to treatment monitoring also in pediatric and adolescent ages. While telemedicine offers convenience and accessibility in healthcare delivery, its application in managing OSAS could be associated with some disadvantages, including limitations in physical examination, access to diagnostic tools, and education and counseling; technology barriers; and privacy concerns. The adoption of a hybrid approach, integrating both in-office and virtual appointments, could effectively meet the needs of children with OSAS. However, more studies are needed to fully assess the effectiveness and safety of telemedicine in the pediatric population.

## 1. Introduction

Telemedicine is defined as the practice of medicine using electronic communications or other forms of digital technology to exchange information allowing the creation of a patient–doctor relationship without the need of a physical meeting [1]. Telemedicine has revolutionized the delivery of medical services, and its history is intricately connected with the evolution of telecommunications [2,3,4,5,6]. 

In recent years, the rapid technological evolution, including advancements in hardware and software (smartphones, tablets, mobile applications, cellular broadband), has reshaped the world of telemedicine [7,8]. Furthermore, the widespread acceptance of smartphones and tablets to perform virtual visits during COVID-19 pandemic restrictions has played a crucial role in the diffusion of telemedicine [7,8]. Telemedicine allows healthcare professionals to remotely diagnose, monitor, and treat patients, transcending geographical barriers. Applications span from virtual consultations and remote patients’ monitoring to telehealth platforms, offering a promising opportunity to enhance healthcare accessibility, especially in underserved areas [9]. Telemedicine can be particularly effective when addressing those in need of continuous assistance, such as patients affected by chronic diseases [10]. This approach reduces the necessity of traveling to reach medical centers, decreasing stress, costs, lost school or workdays, and time spent by families on commutes [11]. In addition, being in a familiar environment means feeling comfortable reducing the white coat syndrome. Telemedicine can be provided in the form of (a) teleconsultation through a virtual video or a telephone visit, (b) telediagnosis, (c) teletherapy, and (d) telemonitoring for short- and long-term follow-up [1]. Communications can be synchronous (such as a video) or asynchronous (like via emails) [12]. Despite telemedicine offering numerous advantages also in pediatric care, some disadvantages must be considered, like technological limitations (i.e., Wi-Fi connectivity, limits in the use of devices, costs), disparities between people in access to digital technologies, lack of physical contact, concerns about data security [13], and legal and regulatory complexities [14]. Telemedicine has been used for sleep disorders including obstructive sleep apnea syndrome (OSAS) since the mid-1990s [15,16]. 

OSAS is defined as a “disorder of breathing during sleep characterized by prolonged partial upper airway obstruction and/or intermittent complete obstruction (obstructive apnea) that disrupts normal ventilation during sleep and normal sleep patterns” [15,16]. OSAS affects 1.2% to 5.7% of children, mainly in the 2- to 6-year age group, with a prevalence increasing alongside the trend of childhood obesity [17]. Sleep is of fundamental importance in pediatric age and plays a crucial role in the development and overall health of children, being closely linked to cognitive functions, learning, emotional well-being, and neuroendocrine regulation [18,19]. Studies have suggested that duration and quality of sleep during childhood can influence cardiovascular health in adulthood and interact with telomere length [18,19]. Telomeres are repetitive DNA sequences and associated proteins that cover the end of chromosomes. They represent a particularly useful biomarker when assessing the health of pediatric populations because disparities in telomere length emerge before the onset of chronic health conditions in adulthood [20]. Nocturnal signs and symptoms of OSAS include habitual sleep snoring, mouth breathing, abnormal thoracic and/or abdominal movements during the night, repeated awakening, restless sleep with pauses in breathing, enuresis, and excessive sweating. During the day, patients with OSAS may report drowsiness, mouth breathing, morning headache, hyperactivity and/or irritability, and poor academic performance [21]. 

OSAS is associated with intermittent oxyhemoglobin desaturations that trigger the production of free radicals in different tissues, leading to persistent systemic inflammation [17]. The persistence of OSAS can result in several behavioral and physical complications. Severe OSAS can cause growth retardation, cardiovascular disease, neurocognitive abnormalities, and behavioral problems and may even contribute to craniofacial malformations and thoracic deformity [17]. Table 1 describes short- and long-term consequences of OSAS.

Three different subtypes of OSAS have been identified: type 1, the commonest in childhood, is associated with adenotonsillar hypertrophy [22,23,24,25]; type 2 is mainly associated with obesity and is more similar to the adult phenotype; and type 3 is associated with craniofacial abnormalities such as achondrodysplasia, Pierre-Robin sequence, macroglossia like in Down syndrome or Beckwith–Wiedemann syndrome, and Prader Willi syndrome [1]. Other at-risk patients for OSAS include children with a history of prematurity, bronchopulmonary dysplasia [23], neuromuscular diseases such as Duchenne muscular dystrophy, and spinal muscular atrophy (SMA) and children with sickle cell disease [24]. No difference in OSAS prevalence has been shown between girls and boys [25].

Type 1 OSAS causes a reduction in the upper airway lumen, as evaluated by the Friedman Grading Scale [22,23,24,25]. The adenoids and tonsils are lymphoid tissues that play a crucial role in immunity, representing the first lymph node stations that come into contact with external agents. The volume of tonsils depends on age and pathological conditions, reaching its peak during the pubertal period [26]. Allergic rhinitis can also trigger an inflammatory process in the nasal mucosa, further exacerbating the limitation of airflow in individuals predisposed to a reduced posterior pharyngeal space due to tonsillar hypertrophy [27]; such obstruction can progress into the oropharynx [28]. A correlation between nasal obstruction and the severity of the clinical picture has been demonstrated through posterior rhinomanometry, highlighting a significant association between the presence of allergic rhinitis and the development of OSAS [29]. 

Obesity represents one of the most important risk factors for the onset of OSA. In type 2 OSAS, the presence of abundant adipose tissue in the neck hinders the upper airways, triggering nocturnal snoring and altering respiratory dynamics [22,23,24,25]. Several studies have demonstrated a connection between the increase in body mass index (BMI) and the incidence of OSAS [30,31]. For instance, Xu et al. reported a direct correlation between BMI standard deviation score (SDS) and the apnea-hypopnea index (AHI) [32,33].

In adult patients with OSAS, telemedicine is helpful both for consultation and diagnosis, the latter obtained through remote recordings of oxygen saturation and further parameters registered with telemonitored respiratory polygraphy or polysomnography. Remote monitoring can be used to follow up the patient and verify adherence to daily treatments including continuous positive airway pressure (CPAP). In children, studies on the role of telemedicine in OSAS are scarce. This narrative review aims to describe the application of telemedicine in children with obstructive sleep apnea syndrome (OSAS), assessing its advantages and disadvantages. 

## 2. Methods

The MEDLINE–PubMed database was searched to collect and select publications from 2003 to 2023. The following combinations of keywords were used: “obstructive sleep apnea syndrome” OR “OSA” OR “OSAS” OR “sleep disordered breathing” AND “telemedicine” AND “children” OR “pediatric” OR “pediatric” OR “adolescent.” We also performed a manual search of the reference lists of the selected studies. The search was limited to English-language journals and full papers only as well as to manuscripts that reported a medical approach on advantages and disadvantages of telemedicine for OSAS in children and adolescents.

## 3. Diagnosis and Telediagnosis of Obstructive Sleep Apnea Syndrome (OSAS)

Telemedicine offers significant support in multiple areas of OSAS diagnosis. It enables remote consultations, allowing patients to discuss symptoms and concerns with sleep specialists without the need for in-person visits [10]. Home sleep studies conducted through telemedicine platforms provide convenient and accessible testing options, facilitating the assessment of sleep patterns and respiratory parameters in the patient’s own environment [10]. Additionally, telemedicine facilitates data analysis and interpretation, allowing sleep medicine specialists to remotely review sleep study results and make accurate diagnoses [11]. It also promotes multidisciplinary collaboration among healthcare professionals involved in OSAS diagnosis, fostering communication and expertise sharing. Furthermore, telemedicine platforms enable patient education and counseling on OSAS diagnosis and management, empowering individuals to actively participate in their care [12]. Overall, telemedicine enhances accessibility, efficiency, and quality of care in the diagnosis of OSAS.

Snoring for three or more months has been associated with an increased risk of OSAS, and, in such cases, a sleep study is recommended [16]. The diagnosis of OSAS requires a thorough clinical and anamnestic evaluation, along with instrumental investigations. Patients with OSAS should be evaluated by a multidisciplinary team made up of a pediatrician, dentist, ear-nose-throat specialist (ENT), neurologist, cardiologist, and psychologist. An essential role is played by the primary care pediatrician, who should receive adequate training on how to formulate the diagnostic suspect of OSAS and on the criteria that should be used to conduct an adequate selection of patients who need to be sent to tertiary care [34]. Table 2 summarizes differences and similarities in the diagnosis and management of sleep-related obstructive respiratory disorders in young children (0–23 months) and older patients (2–18 years).

A primary role is played by the general clinical observation of the child, with particular attention to the “adenoid facies” [35] characterized by underdeveloped cheekbones defining a long face with a flat profile, voluminous and often chapped lips, alteration of the dental structure with upper incisors pushed outward, narrowing of the nostrils, atypical swallowing with a narrow and underdeveloped palate, and deep eye sockets. Attention is also given to the breathing pattern, which in a child affected by OSAS is typically oral, and to the quality of the voice, usually hyponasal or muffled in the presence of tonsillar obstruction. Some details of the physical examination (like the oropharynx) could be missed on virtual visits, and, sometimes, a photo of the oropharynx with the external camera of a smartphone can provide sufficient information for the doctor even if it is not always possible to obtain the best view of the oropharynx due to camera angle or picture clarity, limiting the assessment of the tonsils [36,37]. However, the method of examining the oral cavity/oropharynx via telemedicine appears to be easier to perform during video-linked visits due to improved patient cooperation [38]. Children affected by OSAS may present peculiar anatomical features like increase of the ANB angle (the angle formed between the supraspinale point of the upper jaw, the nasion, and the submentale point of the mandible) and a simultaneous decrease of the SNB angle (the angle formed between the sella turcica, the nasion, and the submentale point of the mandible) [39], an increased height of the inferior-anterior height of the face, reduction of the length of the mandibula [40], or odontostomatological disorders that alter the position of the tongue within the mouth or reduce its space (e.g., cross-bites, surgically corrected cleft palate, high arched palate) [3]. 

A careful anamnestic evaluation is also important, including the use of specific questionnaires, still under study and improvement, but which could help identify patients at risk of OSAS [41]. The Pediatric Sleep Questionnaire (PSQ) is a survey with closed-ended questions, widely used and defined by the guidelines of the Task Force of the European Respiratory Society as a test capable of identifying children with OSAS with AHI > 5 [42]. The Sleep Clinical Score (SCS), based on objective examination, symptoms, and medical history, can be a valuable screening tool to identify patients who may require polysomnography [43]. 

Radiological techniques, including cephalometrics, fluoroscopy, computerized tomography (CT), and magnetic resonance imaging (MRI) [21], are reserved for assessing the airways in individuals with cranial malformations or pre-existing pathological conditions. These methods aim to identify the degree of obstruction, which can determine the type of treatment [44]. 

When OSAS is suspected, nocturnal pulse oximetry or polysomnography should be performed. Nocturnal pulse oximetry is a screening test with good positive predictive value in case of moderate to severe OSAS, but polysomnography remains the gold standard instrumental test for the diagnosis. Several studies [45] attempted to identify children with OSAS using blood oxygen saturation (SpO_2_) measured by nocturnal pulse oximetry. This instrument monitors the child for at least two days in a familiar environment with less discomfort compared to being admitted to the hospital. Although this method presents an accuracy of 86.8%, a sensitivity of 80.0%, and a specificity of 92.1%, it is not considered the gold standard for OSAS diagnosis. The reason is probably attributable to the fact that most children with OSAS have awakenings and sleep fragmentation but little desaturation. On the other hand, polysomnography records various parameters beyond heart rate and blood oxygen saturation such as airflow through the nose and mouth, thoracic and abdominal movements, respiratory rate, electroencephalogram (EEG) to monitor brain activity, electromyography (EMG) to detect muscle movements, and electrooculography (EOG) to record eye movements. It is the only instrument able to distinguish central from peripheral apneas and assess the severity of OSAS based on AHI. AHI corresponds to the number of apneas/hypopneas per hour of sleep and classifies OSAS in mild (AHI 1–4), moderate (AHI 5–9), or severe OSAS (AHI ≥ 10). However, polysomnography is expensive, not available in all centers, and requires hospitalization. 

Theoretically, telemedicine can be used in all aspects of the management of the patient with OSA from diagnosis to therapy since the clinician can review and score polysomnography remotely and during virtual visits. However, the data on the use of telemedicine for the diagnosis of sleep disorders in children are scarce. A position paper from the American Academy of Sleep Medicine (AASM) promotes the use of telemedicine in sleep disorders in adults stating that the specialists involved in the care of these patients should be coordinated to home deliver the care of sleep disorders [46]. 

Although polysomnography is typically performed in specialized sleep labs, studies performed in adult populations report that home sleep studies where sleep technicians remotely and continuously supervise polysomnography have a rate of failure < 10% [47,48]. A pilot study in Brussels utilized a wireless system called Sleepbox^®^, capable of transmitting recorded data during home polysomnography to the central laboratory with excellent data quality [49]. A more simplified home monitor is the home sleep apnea testing (HSAT) that records breathing, blood oxygen saturation, and breathing effort; this is often used for diagnosis of OSAS in adults due to its excellent sensitivity [50]. Currently AASM does not recommend performing HSAT in children [51], but success of HSAT in OSAS diagnosis can be reached in 90% of patients investigated with good compliance of parents and children who can carry out the examination in their own bed [52]. Both the American Academy of Otolaryngology—Head and Neck Surgery [53] and the American Academy of Pediatrics [54] consider this test as an appropriate means to evaluate sleep-disordered breathing in preparation for adenotonsillectomy. In the Tucson Children’s Assessment of Sleep Apnea (TuCASA) study [55], HSAT was performed on 157 children aged 5 to 12 years, with technically acceptable results achieved in 91% of patients on the first night and 97% on the additional night. To be considered acceptable, the registration should be at least 5 h long and include interpretable signals of respiration, EEG, and pulse oximetry. Brockman and colleagues [56] compared HSAT performed at home with HSAT conducted in the hospital in 101 children. In both groups, electrodes were placed by experienced personnel, and the percentage of acceptable recordings was around 93%. The real differentiating factor between conducting HSATs at home or in a hospital setting appears to be the application of sensors by caregivers versus specialized personnel, as demonstrated by Poels et al. in their study [57]. 

Furthermore, HSAT can alleviate the anxiety and discomfort often experienced by patients with behavioral disorders, who typically exhibit reluctance to undergo tests in a hospital setting. The case report described by Donskoy et al. [58] demonstrates, indeed, the utility of HSAT at home in accurately diagnosing and managing sleep-disordered breathing in patients with autism and behavioral disorders, enabling a precise and less invasive evaluation of their sleep in an environment more conducive to their needs. New studies are needed to create a diagnostic algorithm to identify children who do not need CO2 monitoring and therefore are eligible for HSAT. Ideally, these subjects should be subjects who may develop isolated obstructive hypoventilation [51].

## 4. Therapy of Obstructive Sleep Apnea Syndrome (OSAS) and Teletherapy

Telemedicine offers significant assistance in OSAS therapy by facilitating remote patient monitoring, therapy adjustments, and adherence support [17,18,19,20,21]. Patients can receive real-time feedback on their CPAP therapy usage and mask fit, enhancing treatment compliance. Telemedicine platforms enable healthcare providers to remotely assess treatment effectiveness, troubleshoot issues, and make necessary adjustments, improving patient outcomes. Moreover, telemedicine offers convenient access to sleep specialists, educational resources, and behavioral interventions, empowering patients to actively participate in their therapy management. Through telemedicine, OSAS therapy becomes more accessible, personalized, and effective, ultimately improving the quality of patient care.

As previously reported, treatment of OSAS depends on OSAS type, disease severity, age of the child, presence of comorbidities, and coexisting risk factors [34,54] (Table 3). 

### 4.1. Surgical Therapy

Adenotonsillectomy (AT) stands as the gold standard treatment for type 1 OSAS, offering a curative potential for 75–100% of affected patients with a significant improvement in polysomnographic results, which return to normal values in 79% of patients undergoing surgery [59]. In children aged between 2 and 12 years with mild OSAS, it is possible to apply a watchful waiting approach by administering medical therapy. AT remains the preferred approach in patients with moderate-grade OSAS or mild-grade OSAS who present with genetic syndromes or deterioration in the quality of life. However, given the high risk of respiratory complications postoperatively, careful monitoring is essential [60]. A 2020 study by Waliejee et al. on 535 children undergoing otolaryngology surgery for OSAS highlighted the effectiveness of telemedicine [61]. The application of telemedicine, with baseline calls and subsequent video consultations in response to reported symptoms, resulted in a noteworthy reduction in clinic visits. Additionally, this procedure offers a strong safety net for individuals who continue to have issues and require additional research and oversight. In this study, approximately 11 out of 55 children who reported symptoms were able to receive timely video consultations, demonstrating the potential of telemedicine in optimizing postoperative care and minimizing unnecessary in-person visits [61]. In the presence of craniofacial malformations, corrective surgical therapy is indicated, which includes tracheostomy and mandibular osteotomy. A meta-analysis evaluated an overall effectiveness of 87% in reducing AHI in subjects undergoing corrective maxillofacial surgery [62]. Initially, this combined therapeutic approach of mandibular advancement was reserved for dysmorphic individuals with significant mandibular retrusion. Currently, the possibility of isolated mandibular advancement has opened the opportunity to perform such maxillofacial surgery even in eumorphic subjects.

### 4.2. Pharmacological Therapy

Advantages of telemedicine in OSAS pharmacological therapy include enhanced accessibility to healthcare providers, convenient medication management, and improved patient adherence through remote monitoring and support [10,11,12]. Patients can receive timely medication adjustments and counseling, optimizing treatment outcomes. However, disadvantages include limitations in physical examination, inability to perform certain diagnostic tests remotely, and potential challenges in assessing treatment side effects. Additionally, concerns regarding patient privacy, data security, and technological barriers may impact the quality of telemedicine-based pharmacological therapy for OSAS. Despite these drawbacks, telemedicine remains a valuable tool in expanding access to pharmacological interventions for OSAS [10,11,12].

Medical therapy should be prescribed for mild OSAS, when the patient is awaiting surgery [63], or when the patient is unable to undergo surgical intervention [64]. Mild forms eligible for medical therapy are those with an AHI ranging from 1 to 5 events per hour and with nadir SpO2 between 86% and 91% [65]. Anti-inflammatory therapies, such as topical steroids, can be useful [66] by reducing the release of inflammatory mediators that may cause nasal obstruction in individuals with allergic rhinitis and adenoid hypertrophy [67,68]. The use of inhaled steroids, compared to systemic steroids, results in lower systemic absorption by acting locally on the nasal mucosa [69]. The most commonly used corticosteroids are budesonide, fluticasone, and mometasone [67]. Numerous studies have demonstrated clinical improvement after treatment with nasal inhaled corticosteroids compared to the use of placebo, as confirmed by a Cochrane review that highlighted a reduction in AHI values in patients treated with this therapy [66,70,71].

Leukotriene receptor antagonists (LTRA) constitute a class of medications used to reduce inflammation in the upper airways by interfering with the leukotriene pathway [72]. This pathway is involved in the proinflammatory and proliferative process of adenotonsillar hypertrophy in children affected by OSAS. Montelukast, due to its anti-inflammatory action [26], is currently used to treat allergic rhinitis, which often coexists and represents a risk factor for OSAS. Administered for at least 3 months, this medication has shown a 55% improvement in AHI-reducing adenoidal hypertrophy [73].

### 4.3. Continuous Positive Airway Pressure (CPAP)

The use of CPAP as a treatment option for OSAS in infants and children is widely acknowledged [74]. When adenotonsillectomy cannot be performed or if OSAS persists after surgery, CPAP becomes the ideal treatment [75]. The use of CPAP allows for a minimum effective pressure to keep the upper airways open throughout the night, thus preventing apnea events and improving overall sleep quality. In pediatric patients, the standard procedure for determining the therapeutic pressure of CPAP is manual titration during a polysomnography performed at a specialized center [76]. An alternative developed in recent years is the use of auto-titrating CPAP devices (AutoCPAP). These devices adjust the delivered pressure based on various parameters detected by the device itself [77]. This method has been underexplored in pediatric populations due to specific characteristics of this age group like small airways and higher respiratory rates [78]. However, some studies [79,80] have shown that pressures set through manual titration during polysomnography and pressures obtained via AutoCPAP are nearly identical, demonstrating the clinical effectiveness of this method even in pediatric patients.

Remote monitoring improves adherence to treatments, offers to caregivers the opportunity to adjust the settings of the device under the supervision of the clinician, and can eventually contribute to better outcomes [81]. In patients equipped with CPAP, home virtual visits might be useful since the patient facing difficulties with the device can be assisted remotely in the home setting, providing a prompt solution through a simple video call. This not only can resolve the issues but can also avoid stressful visits to the hospital [82].

### 4.4. Weight Loss

In the case of type 2 OSAS, weight loss represents the main treatment. It is recommended that all children engage in at least 60 min of physical activity per day [10]. In-depth studies conducted in both pediatric and adult populations have shown that adopting a healthy lifestyle and concurrent weight loss lead to a 20% improvement in the severity of OSAS [83]. There is, therefore, a direct correlation between obesity and OSAS severity. The primary goal of medical treatment for childhood obesity is to promote a healthy lifestyle by modifying incorrect habits [84]. A recent study conducted by Bala et al. demonstrated excellent compliance and adherence of caregivers to health coaching visits [85]. In this context, mobile apps are widely used to assist in managing obese patients, but there are no studies evaluating the effectiveness of such applications in the pediatric age group. The integration of exercise into computerized games could be a strategy to make regular physical activity less burdensome. Interactive gaming combined with fitness counseling in telemedicine can be an effective tool to promote weight loss and increase physical activity. In a randomized study, patients undergoing exergaming (a video game that involves physical activity) showed a reduction in BMI z-score values and systolic and diastolic blood pressure values, as well as a decrease in blood levels of cholesterol and low-density lipoprotein (LDL) compared to controls. Of note, the adherence to this activity was very high [86]. Telemedicine could be also used as teleconsultation for periodical nutritional and psychological evaluations, allowing a more consistent control of diet and mental well-being [87].

## 5. Telemonitoring

After diagnosis, patients with OSAS have to be monitored and followed over time. Families of children with OSAS report greater financial burdens and impacts on quality of life due to missed hours at school or work [88]. In adult patients, OSAS can be monitored via teleconsultations to explain the therapy (for example, the use of CPAP), assess over time the adherence to treatment, and adapt the settings depending on what is registered and what is referred by patients [89]. Some CPAP devices have wireless built-in connectivity that can transfer data on ventilation to sleep labs [41,42].

The use of telemedicine in OSAS management is described in Figure 1. A multidisciplinary approach plays a crucial role in telemedicine for the management of OSA. In telemedicine settings, a collaborative team consisting of sleep medicine specialists, respiratory therapists, sleep technologists, psychologists, nutritionists, and other healthcare professionals can work together to provide comprehensive care for patients with OSAS. The multidisciplinary approach begins with remote screening and diagnosis, utilizing telemedicine platforms to conduct virtual consultations, sleep studies, and assessments of sleep quality and respiratory parameters. Specialists can collaborate remotely to interpret diagnostic data, formulate treatment plans, and initiate interventions tailored to the individual patient’s needs. This could facilitate ongoing monitoring and follow-up care, allowing patients to receive regular feedback, adjust treatment parameters, and address any concerns or issues that may arise. Through telemedicine platforms, patients can access educational resources, behavioral interventions, and support services to optimize adherence to treatment modalities such as continuous positive airway pressure (CPAP) therapy, oral appliances, or surgical interventions. By leveraging the expertise of a multidisciplinary team and harnessing the capabilities of telemedicine technology, healthcare providers can deliver high-quality, patient-centered care for individuals with OSAS, improving clinical outcomes and enhancing overall quality of life.

## 6. Conclusions

Telemedicine is contributing to the reorganization of healthcare, shifting the focus from hospitals to the community, with innovative care models centered on citizens and facilitating the access to health services [9]. In adulthood, numerous studies have been conducted on the use of telemedicine in OSAS [90]. This approach is applicable at every stage of patient management, from diagnosis to treatment monitoring [91]. Remote monitoring devices, such as wearables and at-home sleep monitoring equipment, enable diagnosis and continuous monitoring of patients with OSAS [92]. The collected data can be transmitted to healthcare professionals for in-depth analysis, including the transmission of data from a peripheral hospital to a central hub. Telemedicine can also promote patients’ education about OSAS and optimize adherence to therapies, such as the use of CPAP [89,93]. Telemedicine can also support lifestyle modification, such as weight loss or promotion of physical activity. While telemedicine offers convenience and accessibility in healthcare delivery, its application in managing OSAS could be associated with some disadvantages, including limitations in physical examination, access to diagnostic tools, and education and counseling; technology barriers; and privacy concerns. Addressing these challenges is essential to optimize the delivery of telemedicine services and ensure comprehensive care for patients with OSA. Although home sleep studies are not recommended in patients younger than 18 years of age in the diagnosis of OSAS [51], telemedicine represents a promising option for managing OSAS from diagnosis to monitoring also in children. To date, the clinical use of HSATs in children is not recommended due to insufficient evidence, but some studies suggested that HSAT may be technically feasible even in younger patients. The adoption of a hybrid approach, integrating both in-office and virtual appointments, could effectively meet the needs of children with OSAS. The effectiveness of this hybrid model has been demonstrated in a fragile group of patients as those scheduled to receive a clinical electrophysiology evaluation [94]. However, more studies are needed to fully assess the effectiveness and safety of telemedicine in this population as well as the cost-effectiveness of this approach. In addition, as suggested for pediatric urology [95], the metaverse, i.e., an emerging resource in healthcare resulting from the integration of virtual and physical reality, could allow to empower pediatric OSAS experts, patients, and their families.

## Figures and Tables

**Figure 1 jcm-13-02108-f001:**
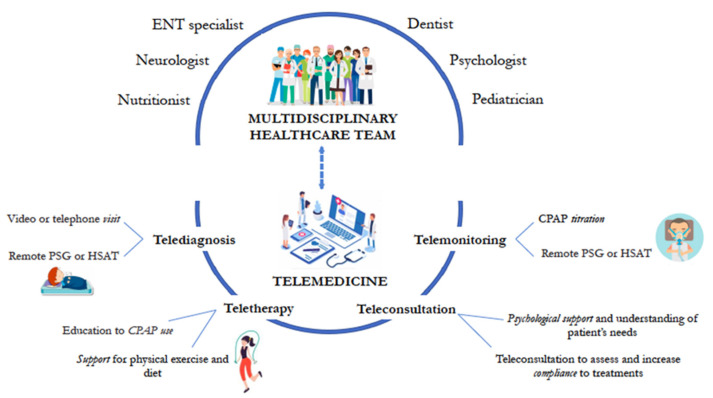
Telemedicine in the management of obstructive sleep apnea syndrome (OSAS). CPAP, continuous positive airway pressure; ENT, ear-nose-throat; HSAT, home sleep apnea testing; PSG, polysomnography.

**Table 1 jcm-13-02108-t001:** Short- and long-term consequences of obstructive sleep apnea syndrome (OSAS).

Short-Term Consequences	Long-Term Consequences
Snoring	Behavior problems (hyperactivity and/or irritability)
Abnormal chest/abdomen motion	Excessive daytime sleepiness
Excessive sweating	Development delay
Cyanosis during sleep	Poor academic performance
Disturbed sleep (e.g., repeated awakening or position changes)	Enuresis
Nasal obstruction	Cardio-respiratory complications (e.g., cor pulmonale)
Oral breathing	Craniofacial deformation
Drowsiness	Thoracic deformities
Morning headache	

**Table 2 jcm-13-02108-t002:** Differences and similarities in the diagnosis and management of sleep-related obstructive respiratory disorders in young children (0–23 months) and older patients (2–18 years).

Diagnosis	Patients 0–23 Months	Patients 2–18 Years
Symptoms of upper airway obstruction present in both wakefulness and sleep	Yes	No
Adenotonsillar hypertrophy and obesity as a cause of sleep-related obstructive respiratory disorders	Yes, but uncommon	Yes
Syndromes, congenital anomalies as a cause of sleep-related obstructive respiratory disorders	Yes	Yes
Feeding difficulties and poor growth can coexist with OSA	Yes	No
Pulmonary hypertension can complicate OSA	Yes	Yes
Polysomnography as the gold standard for OSA	Yes	Yes
Endoscopy useful for assessing upper airway collapse	Yes	No
Management	Yes	Yes
Adenotonsillectomy is the most useful treatment	Yes	Yes
Non-invasive ventilation is often used as a first treatment for dynamic airway collapse	Yes	No
Effective orthodontic appliances in cases of OSA with retrognathia and malocclusion	No	Yes
Patients with complex conditions to be treated as a priority	Yes	Yes
Follow-up after surgery should detect persistent OSA	Yes	Yes
Patients on non-invasive ventilation undergo annual nocturnal saturation monitoring	Yes	Yes

**Table 3 jcm-13-02108-t003:** Treatment of OSAS depending on OSAS type.

	Medical Therapy	Surgical Therapy
Type 1 OSAS	- CPAPNasal steroids and leukotriene receptor antagonists	Adenotonsillectomy
Type 2 OSAS	- Weight loss - CPAP - Nasal steroids and leukotriene receptor antagonists	Bariatric surgery
Type 3 OSAS	- Rapid maxillary expansion - CPAP - Nasal steroids and leukotriene receptor antagonists	Craniofacial surgery

CPAP, continuous positive airway pressure.

## Data Availability

Not applicable.

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
