# Peer review of "The Role of Telemedicine in Children with Obstructive Sleep Apnea Syndrome (OSAS): A Review of the Literature"

_jcm, 2024, doi:10.3390/jcm13072108_

Round 1

Reviewer 1 Report

Comments and Suggestions for Authors

The authors should be congratulated for their work. The role of Telemedicine is growing nowadays, due to the Covid-19 outbreak that has severely undermined access to hospitals and territory health services (PMID= 36576476 ,37530670, 35565003) . Specifically, as the authors stated the use of Telemedicine in OSAS children has scarce evidence and in consequence, it is worthy to be explored. Specifically, their narrative review aimed to address this knowledge gap. Their work sounds exhaustively well done. They explored the topic from the diagnosis to the follow-up. Here, they showed that telemedicine is applicable at every stage of patient management, both in pediatric and adolescent patients. Furthermore, the adoption of a hybrid approach (live and virtual appointments), could effectively meet the needs of children with OSAS. Carefully, they suggested that the above phenomena are observed, but to assess the effectiveness and safety of telemedicine in the pediatric population further studies, with a different design are needed. 

I suggest adding a Supplementary table in which the keywords should be acknowledged. It could be an additional item useful to understand the paucity of data regarding telemedicine in OSAS. Moreover, the authors should definitely describe this pioneering paper on the potential creation of a metaverse that could apply to pediatric urology as well as other disciplines (DOI: https://doi.org/10.3390/surgeries4030033). It would be interesting to know their opinion and the future perspective in ENT field. 

Author Response

The authors should be congratulated for their work. The role of Telemedicine is growing nowadays, due to the Covid-19 outbreak that has severely undermined access to hospitals and territory health services (PMID= 36576476 ,37530670, 35565003) . Specifically, as the authors stated the use of Telemedicine in OSAS children has scarce evidence and in consequence, it is worthy to be explored. Specifically, their narrative review aimed to address this knowledge gap. Their work sounds exhaustively well done. They explored the topic from the diagnosis to the follow-up. Here, they showed that telemedicine is applicable at every stage of patient management, both in pediatric and adolescent patients. Furthermore, the adoption of a hybrid approach (live and virtual appointments), could effectively meet the needs of children with OSAS. Carefully, they suggested that the above phenomena are observed, but to assess the effectiveness and safety of telemedicine in the pediatric population further studies, with a different design are needed.

Re: Thank you very much for your positive evaluation. We revised the manuscript according to your suggestions and those received from other reviewers.

I suggest adding a Supplementary table in which the keywords should be acknowledged. It could be an additional item useful to understand the paucity of data regarding telemedicine in OSAS. Moreover, the authors should definitely describe this pioneering paper on the potential creation of a metaverse that could apply to pediatric urology as well as other disciplines (DOI: https://doi.org/10.3390/surgeries4030033). It would be interesting to know their opinion and the future perspective in ENT field.

Re: Keywords are included in the text (p. 2). The suggested manuscript has been mentioned for future studies (p. 11) and the reference added (p. 16).

Reviewer 2 Report

Comments and Suggestions for Authors

Congratulations to the authors for the interesting idea of the manuscript as telemedicine is gaining a leading role in medicine and it is necessary to study every aspects of it. The paper will be fine after major revisions; I have many comments on it and I am sure that authors will appropriately answer to all of them.

Can you include any study that provided info regarding the cost-effectiveness of this kind of telemedicine in this setting of patients, not just in OSAS disease but in pediatric patient setting.

You should add some table in order to make more easily readable your paper

In order to enrich your bibliography and strengthen the feasibility of telemedicine in every aspects of medicine I strongly suggest you to consider adding: “The Feasibility, Effectiveness and Acceptance of Virtual Visits as Compared to In-Person Visits among Clinical Electrophysiology Patients during the COVID-19 Pandemic. J Clin Med. 2023 Jan 12;12(2):620. doi: 10.3390/jcm12020620. PMID: 36675547; PMCID: PMC9865180.” As represents a valuable example of telemedicine in a fragile category of patients.

Stress the role of multidisciplinary approach in telemedicine (Physicians, technicians, etc)

I think that you should expand the telemedicine section, for example explaining figure 2 in a dedicated paragraph.

Comments on the Quality of English Language

minor english correction are needed

Author Response

Congratulations to the authors for the interesting idea of the manuscript as telemedicine is gaining a leading role in medicine and it is necessary to study every aspects of it. The paper will be fine after major revisions; I have many comments on it and I am sure that authors will appropriately answer to all of them.

Re: Thank you for your suggestions. We revised the manuscript according to your comments and recommendations.

Can you include any study that provided info regarding the cost-effectiveness of this kind of telemedicine in this setting of patients, not just in OSAS disease but in pediatric patient setting.

Re: Ref. 9 summarizes this unmet needs and we have highlighted it among priorities for future studies (p. 11).

You should add some table in order to make more easily readable your paper

Re: Our manuscript include two Tables and two Figures, we think that it is appropriate and more Tables could create confusion.

In order to enrich your bibliography and strengthen the feasibility of telemedicine in every aspects of medicine I strongly suggest you to consider adding: “The Feasibility, Effectiveness and Acceptance of Virtual Visits as Compared to In-Person Visits among Clinical Electrophysiology Patients during the COVID-19 Pandemic. J Clin Med. 2023 Jan 12;12(2):620. doi: 10.3390/jcm12020620. PMID: 36675547; PMCID: PMC9865180.” As represents a valuable example of telemedicine in a fragile category of patients.

Re: Added (pp. 11 and 16).

Stress the role of multidisciplinary approach in telemedicine (Physicians, technicians, etc). I think that you should expand the telemedicine section, for example explaining figure 2 in a dedicated paragraph.

Re: Included (p. 10).

Reviewer 3 Report

Comments and Suggestions for Authors

Comments to the authors:

Thank you for inviting me to review the article entitled “The role of telemedicine in children with OSAS”. 

This is a review on the current state of the art of telemedicine used for the management of children with OSAS. The study aims at summarizing advantages and disadvantages of telemedicine for OSAS in pediatric population. However, the result section (which should be the one following the indication of the lit search) starts with a description of OSAS, risk factors, and pathophysiology. This makes little sense. I suggest instead that the authors incorporate and summarize this in the introduction, as the introduction should contain information on telemedicine and also on OSAS. Subsequently, they can just dedicate the result section to the advantages and disadvantages of telemedicine for OSAS, as explicated in their study aim.  

Moreover, while it is particularly clear which advantages telemedicine has towards management of OSA process, it is not clear what the authors identify as disadvantages (unless they refer to HSAT as a disadvantage). It would be useful to add a table that can summarize advantages and disadvantages of telemedicine at each stage of the process of OSA care. 

Other minor points are here expressed: 

Title: this should contain the type of study. So, please add “A review of the literature” or anything similar to it. 

Abstract: this is good and concise. If word permits, the authors could also list some of the advantages and disadvantages as a result of their search of the literature. 

Introduction: 

I would not take such a broach approach to the topic. For this reason, I would omit the paragraphs from line 35 to 59. It does not make much sense right now to talk about the history of telemedicine, as it has not become integral part of healthcare. 

Lines 89-95: this should be placed in the method section. Please, create a new section of methods and move this part to it. Also, please add that the search was limited to summarize advantages and disadvantages of telemedicine for OSAS> 

Line 109-110: “Studies have suggested 109 that duration and quality of sleep during childhood can influence cardiovascular health 110 in adulthood and interact with telomere length.” This sentence requires a citation. 

Line 120-122: this also requires a citation. 

Table 1: I would remove “apneas” from short-term consequences, as this is indeed OSAS.

Figure 1 does not make sense. In case, figure 1 could depict two clinical situations where, according to the value of ANB and SNB, OSAS may be more severe or more prevalent. But a figure 1 just displaying a cephalometric landmark does not fit the current study.

Section 3. Line 157. This should be proceeded by a paragraph where the authors indicate in which areas of the process of management OSAS telemedicine can help, for example the one displayed in Figure 2 (which to my personal opinion should be moved earlier in the manuscript to present the results). Then, section 3 makes sense because it is part of the screening and identification. It may also make sense here to add a table with all the features of the clinical examination that can aid in the identification and rise red flag for risk of OSA (in addition to the few ones mentioned here by the authors, e.g., adenoid facies, hypertrophy of tonsils, etc).

Then, from line 208, the authors start talking about the diagnosis, which could be isolated in a new section. 

The section from lines 234 to lines 266 present the advantages and disadvantages of HSAT. Do the authors consider HSAT a telemedicine tool? This needs to be explicated, so that this section can have a logic explanation in line with the aim of the review. 

Similarly, the section 4 should clearly state why and how telemedicine can be useful as an aid during the management. Otherwise, the reader starts reading about all the therapeutical approach which do not fit with the aim of the study. Please, just rephrase and add an introductory paragraph where the authors state the importance of long-distance monitoring in case of management with surgeries etc. 

Please, add citation to Table 2 to support these data. 

Aslo section 4.2 Pharmacological therapy does not fit much if the aim of the review is to summarize advantages and disadvantages. The data here expressed needs to be presented in light with the aim of the study. 

Author Response

Thank you for inviting me to review the article entitled “The role of telemedicine in children with OSAS”.

Re: Thank you for your suggestions. We revised the manuscript according to your recommendations.

This is a review on the current state of the art of telemedicine used for the management of children with OSAS. The study aims at summarizing advantages and disadvantages of telemedicine for OSAS in pediatric population. However, the result section (which should be the one following the indication of the lit search) starts with a description of OSAS, risk factors, and pathophysiology. This makes little sense. I suggest instead that the authors incorporate and summarize this in the introduction, as the introduction should contain information on telemedicine and also on OSAS. Subsequently, they can just dedicate the result section to the advantages and disadvantages of telemedicine for OSAS, as explicated in their study aim. 

Re: Revised as suggested (pp. 2-4).

Moreover, while it is particularly clear which advantages telemedicine has towards management of OSA process, it is not clear what the authors identify as disadvantages (unless they refer to HSAT as a disadvantage). It would be useful to add a table that can summarize advantages and disadvantages of telemedicine at each stage of the process of OSA care.

Re: Possible disadvantages have been included in the text (p. 11).

Other minor points are here expressed:

Title: this should contain the type of study. So, please add “A review of the literature” or anything similar to it.

Re: Revised as suggested (p. 1).

 Abstract: this is good and concise. If word permits, the authors could also list some of the advantages and disadvantages as a result of their search of the literature.

Re: Disadvantages have been added in the Abstract.

Introduction:

I would not take such a broach approach to the topic. For this reason, I would omit the paragraphs from line 35 to 59. It does not make much sense right now to talk about the history of telemedicine, as it has not become integral part of healthcare.

Re: Done (pp. 1-2).

Lines 89-95: this should be placed in the method section. Please, create a new section of methods and move this part to it. Also, please add that the search was limited to summarize advantages and disadvantages of telemedicine for OSAS>

Re: Done (p. 4).

Line 109-110: “Studies have suggested 109 that duration and quality of sleep during childhood can influence cardiovascular health 110 in adulthood and interact with telomere length.” This sentence requires a citation.

Re: Added (p. 2).

Line 120-122: this also requires a citation.

Re: Added (p. 2).

Table 1: I would remove “apneas” from short-term consequences, as this is indeed OSAS.

Re: Revised (p. 2).

Figure 1 does not make sense. In case, figure 1 could depict two clinical situations where, according to the value of ANB and SNB, OSAS may be more severe or more prevalent. But a figure 1 just displaying a cephalometric landmark does not fit the current study.

Re: Figure 1 has been deleted as requested.

Section 3. Line 157. This should be proceeded by a paragraph where the authors indicate in which areas of the process of management OSAS telemedicine can help, for example the one displayed in Figure 2 (which to my personal opinion should be moved earlier in the manuscript to present the results). Then, section 3 makes sense because it is part of the screening and identification. It may also make sense here to add a table with all the features of the clinical examination that can aid in the identification and rise red flag for risk of OSA (in addition to the few ones mentioned here by the authors, e.g., adenoid facies, hypertrophy of tonsils, etc).

Re: The requested paragraph has been added (p. 4). A Table on clinical findings has been included as suggested (pp. 4-5).

Then, from line 208, the authors start talking about the diagnosis, which could be isolated in a new section.

The section from lines 234 to lines 266 present the advantages and disadvantages of HSAT. Do the authors consider HSAT a telemedicine tool? This needs to be explicated, so that this section can have a logic explanation in line with the aim of the review.

Re: Disadvantages are reported in the Conclusions (p. 11).

Similarly, the section 4 should clearly state why and how telemedicine can be useful as an aid during the management. Otherwise, the reader starts reading about all the therapeutical approach which do not fit with the aim of the study. Please, just rephrase and add an introductory paragraph where the authors state the importance of long-distance monitoring in case of management with surgeries etc.

Re: A paragraph has been added as suggested (pp. 7-8).

Please, add citation to Table 2 to support these data.

Re: References are mentioned in the text (p. 8). The Table is original.

Aslo section 4.2 Pharmacological therapy does not fit much if the aim of the review is to summarize advantages and disadvantages. The data here expressed needs to be presented in light with the aim of the study.

Re: A paragraph has been added as requested (p. 9).

Round 2

Reviewer 1 Report

Comments and Suggestions for Authors

The authors addressed properly my comments. 

Reviewer 2 Report

Comments and Suggestions for Authors

Congratulations to authors as the newest version of manuscript is really improved

Reviewer 3 Report

Comments and Suggestions for Authors

The authors have now provided an improved version of the manuscript, which is more logic, comprehensive, and in the scope of the aim of the study. I am satisfied with the current version.